# Synergy-Based Sensor Reduction for Recording the Whole Hand Kinematics

**DOI:** 10.3390/s21041049

**Published:** 2021-02-04

**Authors:** Néstor J. Jarque-Bou, Joaquín L. Sancho-Bru, Margarita Vergara

**Affiliations:** Department of Mechanical Engineering and Construction, Universitat Jaume I, E12071 Castellón, Spain; sancho@uji.es (J.L.S.-B.); vergara@uji.es (M.V.)

**Keywords:** hand kinematics, principal component analysis, synergies, dimensionality reduction, joint angles estimation

## Abstract

Simultaneous measurement of the kinematics of all hand segments is cumbersome due to sensor placement constraints, occlusions, and environmental disturbances. The aim of this study is to reduce the number of sensors required by using kinematic synergies, which are considered the basic building blocks underlying hand motions. Synergies were identified from the public KIN-MUS UJI database (22 subjects, 26 representative daily activities). Ten synergies per subject were extracted as the principal components explaining at least 95% of the total variance of the angles recorded across all tasks. The 220 resulting synergies were clustered, and candidate angles for estimating the remaining angles were obtained from these groups. Different combinations of candidates were tested and the one providing the lowest error was selected, its goodness being evaluated against kinematic data from another dataset (KINE-ADL BE-UJI). Consequently, the original 16 joint angles were reduced to eight: carpometacarpal flexion and abduction of thumb, metacarpophalangeal and interphalangeal flexion of thumb, proximal interphalangeal flexion of index and ring fingers, metacarpophalangeal flexion of ring finger, and palmar arch. Average estimation errors across joints were below 10% of the range of motion of each joint angle for all the activities. Across activities, errors ranged between 3.1% and 16.8%**.**

## 1. Introduction

The human hand is a complex biomechanical system, with an intricate kinematics provided by 19 joints, some with various degrees of freedom (DoF). This complexity is key for the versatility of the hand, enabling a large number of activities to be performed with a high level of precision. The measurement of the kinematics of the hand can provide useful information in different fields, such as clinical practice, prosthesis control, teleoperation, or virtual reality. However, the high number of DoF makes both hand movement measurement and its subsequent analysis difficult. 

The recording of hand kinematics can be carried out using different devices that employ different motion capture technologies [1]: Electrogoniometers [2], instrumented gloves [3,4,5], optical tracking systems [6,7,8,9] or inertial sensors that incorporate magnetometers, accelerometers, and gyroscopes [10,11,12,13]. Electrogoniometers are commonly used in clinical practice to measure the range of motion (RoM) of joints [14]. However, due to their size, they can only be used to record a few hand joints simultaneously and are invasive. Optical tracking systems (videogrammetry) is the most widely used technique in biomechanics [15] and it is often considered the gold standard [16]. The cameras are calibrated to track the movement of the markers, which can be reflective (passive markers) or self-illuminated (active markers). Four different skin marker sets are commonly used to record hand motion [16,17,18]: “one marker per segment”, where one marker is positioned at each joint; “two markers per segment”, with markers at the distal and proximal heads of each segment; and “three markers per segment”, where markers are placed forming a triangle on each segment. Methods with one or two markers are prone to larger skin movements, because the joint heads of the fingers have many wrinkles in the skin [19,20]. However, those with three markers per segment are more time-consuming (high placement times and subsequent follow-up processing) and prone to occlusions (by overlapping of markers, body segments—mainly the hand—and the manipulated objects, especially in the case of activities of daily life (ADL)). In recent years, inertial sensors have been used as an alternative to measure kinematics. These systems are easy to transport and manipulate and measure captured kinematic information with high accuracy [21]. However, they present limitations such as the need for an orientation algorithm (such as a Kalman filter), drift, noise, temperature influence, and possible disturbances of the magnetic field [11,22]. In addition, they are affected by movements of the skin and their size limits their use for measuring hand kinematics. Instrumented gloves are quite common when recording the continuous movement of all hand joints because they have no occlusion problems and no spatial environmental constraints, although wearing the gloves affects hand skills [23]. Furthermore, due to placement constraints some of the glove sensors have a non-linear relationship with anatomical angles, therefore requiring specific calibration protocols [24,25] to address this issue, which extends the duration of the test. In summary, the simultaneous measurement of all hand segments is cumbersome due to environment disturbances, occlusions, and placement and processing times. Furthermore, the higher the number of sensors is, the more expensive the equipment will be, and not all laboratories or clinicians can afford this. Therefore, reducing the number of hand joints to be recorded, by estimating some joint angles based on the values of others, would reduce occlusion problems, placement times, and post-follow-up processing times, as well as the investment required.

The concept of hand kinematic synergies [26] is being more extensively used (especially in robotics and prosthetics) to make the analysis of the simultaneous movement of all hand joints affordable, as hand movements are coordinated because of mechanical and neurological couplings [27,28]. These synergies are suggested as a way to represent the basic building blocks underlying natural hand motions that can be used to reduce the dimensionality of hand kinematics [29,30]. Although there are other methods to compute synergies, Principal component analysis (PCA) is the most used statistical method for obtaining kinematic synergies because it allows sparse synergies to be obtained [31,32,33,34,35,36,37,38]. Previous studies have shown that a few linear combinations of the hand joint movements (principal components, PCs) could account for most of the variance in the original set of hand postures [31,32,33,35,36]. Lower order PCs correspond to the gross motion of the hand (the more “basic” patterns of finger motion) and are similar across studies, that is, hand opening/closing caused by motion at all metacarpal-phalangeal and/or proximal-interphalangeal joints. Higher order PCs correspond to more subtle motions (more “specific” patterns) and provide additional information about the object to be manipulated [36,39]; thus, higher PCs differ depending on the tasks or grasps considered [31]. Therefore, given the wide variety of ADL humans can perform, a proper selection of a limited set of representative tasks is needed to obtain representative kinematic synergies. The observations from previous studies led to the suggestion that hand posture control might be implemented by combining postural synergies ranging from those responsible for the general shape of the hand (lower PCs) to those responsible for subtler kinematic adjustments [27]. The same reasoning can be applied to the recording of hand kinematics. The kinematic synergies could be applied to obtain the whole hand kinematics from the recording of only a few joint angles, by estimating the remaining angles from the coordination established by those synergies. The number of PCs considered in this procedure would depend on the level of accuracy required. Lower PCs would be enough when estimating the whole hand posture in fields where precision is not so important (as in the case of virtual reality), therefore requiring the recording of only a few DoF. Conversely, a high number of PCs would be required to estimate the hand posture in fields like teleoperation, where higher precision is necessary, therefore requiring the recording of more DoF. However, few studies have used kinematic synergies to simplify the recording of the whole hand. In particular, Ciotti and colleagues [40,41] used five sensors for hand pose recognition, although based on kinematic synergies that were not representative of the global population or of activities of daily living (synergies extracted from only one subject, grasping imagined objects). Furthermore, important hand motions were disregarded, as palmar arch was not measured and because the way methods were applied underestimated movements of joints with smaller ranges of motion such as finger abductions.

The aim of this study is to identify the minimum set of DoF of the hand needed to record the whole hand kinematics through the use of representative kinematic synergies. Hand synergies were identified for a set of 26 representative ADL [3]. Different combinations of DoF were tested as representative of the hand kinematics, and the errors caused by their use in estimating the rest of the DoF are presented. Finally, the solution with the lowest error was selected to evaluate the goodness of the method, and the implications of using these reduced sets of DoF are discussed.

## 2. Materials and Methods

In order to select the set of DoF that best represent the hand kinematics, the following procedure was adopted, which will be described in greater detail later:(1)Hand kinematic synergies extraction: For each subject in the KIN-MUS UJI database [3], subject-specific kinematic synergies were extracted by applying a PCA to the kinematic data recorded during the performance of the 26 ADL in the database.(2)Synergy clustering and selection of candidate DoF for each synergy: Hierarchical Clustering was used to group extracted synergies that are similar among subjects, and one or more representative joint angles were chosen for each resulting synergy as candidate DoF.(3)Selection of the best combination of angles: The joint angles that were not selected as representative were estimated from different combinations of those that were selected. Root mean square errors (RMSE) of the estimated joint angles were computed and the best combination of representative joint angles was selected.(4)Goodness of the method: Using the joint angles selected in the previous step, the joint angles recorded in another kinematic database (33 complex ADL from 20 subjects of KINE-ADL BE-UJI database) were estimated, and RMSE were computed.

### 2.1. Hand Kinematic Synergies Extraction

#### 2.1.1. Experiment A 

Kinematic data from the publicly available KIN-MUS UJI database [3] was used for the extraction of synergies. This section briefly describes this experiment (see [3] for more details). Twenty-two right-handed subjects participated in the experiment, whose mean ± SD age was 35 ± 9 years. The criteria used to select subjects were gender parity in the overall data, age between 20 and 65 years, and no declared upper limb pathologies. Subjects performed 26 representative ADL (Figure 1): 20 activities adapted from the Sollerman Hand Function Test (to ensure their repeatability and to favor their standardization), and six additional activities (A10, A15, A19, A24, A25, A26) that were added based on the percentage of use of the commonest grasps in ADL [42]. In order to foster repeatability, precise instructions for each task were provided and each ADL started and ended with the body and arms in the same posture (arms and hands relaxed at the side of the body when subjects were standing, or arms and hands resting in a relaxed position on the table when they were sitting). They performed each activity once and all subjects did them in the same order.

#### 2.1.2. Kinematics Acquisition

Kinematic data of the right hand was acquired using a CyberGlove (CyberGlove Systems LLC; San Jose, CA, USA) instrumented glove connected to a laptop at 100 Hz. This glove has 18 strain gauges that allow the anatomical angles of the underlying joints to be determined. Right-hand kinematics was recorded while performing these ADL, following a validated calibration protocol that includes some non-linear corrections to obtain anatomical angles [25]. Sixteen joint angles were recorded: flexion of the metacarpophalangeal (MCP) joints of all the fingers and thumb, interphalangeal (IP) joint of the thumb, and proximal interphalangeal (PIP) joints of the fingers; flexion and abduction of the thumb carpometacarpal (CMC) joint; relative abduction between fingers (index-middle; middle-ring; and ring-little); and palmar arch (PalmArch). A reference posture (hand resting flat on a table with fingers and thumbs close together, middle fingers aligned with forearms) was recorded before recording the hand kinematics while performing the selected ADL, and was considered zero for all the joint angles [25]. The recorded joint angles were filtered by a 2nd-order 2-way low-pass Butterworth filter with a cut-off frequency of 5 Hz [43,44], and the initial and final frames of each recording during which the hands remained static were trimmed.

#### 2.1.3. Synergies Extraction

A PCA was applied to the whole kinematic data of each subject, including all the frames for all the ADL. In order for all the ADL to weigh the same in the analyses, the number of frames of each record was rescaled to 1000 frames per ADL [31]. Each PCA matrix input was composed of an ensemble of 16 joint angle time-profiles (1000 frames) for the 26 ADL (matrix dimension 16 × 26,000). Prior to computation of the PCs, the joint angles were normalized by rescaling them to unit variance (mean = 0 and SD = 1 for each DoF) in order to prevent the first PCs from reflecting the joint angles with the largest amplitudes [45]. For each subject, the first PCs explaining at least 95% of the total variance were extracted, and Varimax rotation was applied to obtain more sparse synergies [38]. The PCs thus obtained represented the new variables that substituted the original ones and each PC contained the correlation coefficient (CC) of the new variable with the original variables (joint angles). Finally, the number of the PCs (synergies) extracted and the variance explained for each subject are presented.

### 2.2. Synergy Clustering and Selection of Representative DoF for Each Synergy

All the extracted PCs of all the subjects were included in a cluster analysis. Conglomerate or hierarchical clustering analysis [46] is a multivariate technique that allows elements to be classified into groups or clusters, so that each element is very similar to those in its own conglomerate according to several specific selection criteria. In this case, the angle between PCs [43] was used as the pairwise distance to generate the groups. The angles between PCs were used as input to a hierarchical complete-linkage clustering procedure (or farthest neighbor clustering). The PCs were hierarchically grouped depending on their similarity represented by their pairwise distance (angle between PCs), and the results were presented in a dendrogram. The cut-off distance defining the number of groups was chosen by looking for the minimum number of clusters that ensured no cluster contains more than one synergy from the same subject. Each resulting cluster represents a different kinematic synergy, which was finally described by: (1) the averaged PC (the average of the CC with each original joint angle of all the PCs in the cluster) and a short description of the implicit coordination; (2) the percentage of subjects presenting the synergy; and (3) the mean variance explained. 

One or more representative joint angles were chosen for each resulting synergy from the hierarchical cluster, distinguishing two different cases depending on the values of the CC [47] in the averaged PC representing the cluster:If the averaged PC represents the predominant motion of only one DoF (CC > 0.8 for only one DoF, and CC < 0.3 for all other DoF), that independent DoF was selected as representative.Otherwise, the averaged PC represents a coordinated motion of different DoF, and all DoF with CC > 0.4 were considered as candidates to be representative.

Finally, with the independent joint angles selected in case 1 and with the different combinations of candidates selected in case 2, all other non-representative joint angles were estimated as explained in the next section.

### 2.3. Selection of the Best Combination of Angles

In order to estimate all the non-representative joint angles, the estimated physiological angle for a given frame *i* ($Ang_{est_{i}}$) was proposed to be expressed as a linear combination of the representative angles recorded for the same frame *i* ($Ang_{rep_{k,i}\;}$), in accordance with Equation (1), where *n* is the total number of representative angles:(1)$$\;Ang_{est_{i}}=\;Intercept+\overset{n}{\underset{k=1}{\sum }}Ang_{rep_{k,i}\;\;}*\;x\left(k\right)$$

For each combination of representative joint angles tested, a least-squares fitting procedure was applied to obtain the intercept and coefficients x(k) by using the MATLAB function ‘lsqcurvefit’, which finds coefficients that minimize the sum of the squared differences (Equation (2)) between the estimated angles ($Ang_{est_{i}}$) and the recorded angles ($Ang_{rec_{i}}$) across all frames (22 subjects × 26 ADL × 1000 frames = 572,000 frames altogether): (2)$$\underset{x}{\min}\overset{572000}{\underset{i=1}{\sum }}{\left(Ang_{est_{i}}-Ang_{rec_{i}}\;\right)}^{2}$$

For each non-representative joint angle, once the intercept and coefficients x(k) had been obtained for each combination of representative joint angles tested, the RMSEs between the estimated joint angles and the recorded ones across all frames were obtained. The best combination of representative joint angles (with the lowest averaged RMSE across non-representative joint angles) was finally proposed for the estimation of physiological angles, and its goodness was checked as detailed in the next section.

### 2.4. Evaluation of the Goodness of the Method

#### 2.4.1. Experiment B

Data from the publicly available KINE-ADL BE-UJI database [4] were used to evaluate the goodness of the method against new kinematic data. In this case, the database contains the hand kinematics of both hands of 20 healthy subjects, whose mean ± SD age was 38 ± 9.5 years, during the performance of a wide variety of ADLs (Figure 2) related to eating and food preparation. The criteria used to select subjects were gender parity in the overall data, age between 20 and 65 years, and no declared upper limb pathologies. The hand joint angles were recorded with the same instrumented glove used in Experiment A, and applying the same protocol to obtain anatomical angles and the same filtering. Unlike in Experiment A, the subjects had more freedom to carry out the activities, which were more complex and varied, since each of them involved several actions. For example, the task ‘pouring and drinking milk’ involved opening the milk carton, pouring the milk from the carton into the cup, closing the carton, and drinking from the cup. Only right-hand kinematics data were used for the evaluation of the method. 

#### 2.4.2. Evaluation of the Goodness

By using Equation (1) and the intercept and coefficients x(k) obtained in the previous step, joint angles in the non-representative DoF were estimated for each frame in Experiment B (33 activities and the 20 subjects—1,485,004 frames altogether). Again, RMSE between estimated and recorded joint angles were obtained as:Global RMSE errors (across all frames and subjects, thus one RMSE value per joint)RMSE per ADL (across subjects and frames)

RMSE values are also presented as a percentage of the RoM of each DoF (difference between maximum and minimum values recorded in both experiments together) in order to consider the variability in the ranges of motion of the different hand joints. Finally, comparison of the temporal evolution of the estimated angles and the recorded ones are presented for two trials from Experiment B: the ones with the lowest and the highest RMSE values.

## 3. Results

### 3.1. Hand Kinematic Synergies Extraction

Ten PCs were extracted in all the subjects. The total amount of variance explained per subject is summarized in Figure 3 by portraying the subject distribution in a histogram. The histogram of variance explained for each subject shows the data at 96.48%  ±  1.2%.

### 3.2. Synergy Clustering and Selection of Representative DoF for Each Synergy

Figure 4 shows the dendrogram obtained from the hierarchical clustering algorithm of the whole dataset of PCs (220 extracted PCs = 22 subjects × 10 PCs). The different possible groupings, depending on the similarity between PCs according to their pairwise distances, have been plotted in colors. In this case, the minimum number of groups that ensures that no cluster contains more than one synergy from the same subject corresponds to a cut-off of 60 degrees as the pairwise distance (dotted line in Figure 4), which yielded 14 groups of synergies.

Table 1 describes the groups obtained and the predominant DoF of each group (those DoF with CC > 0.4). Six groups (2, 3, 4, 5, 7 and 9) showed a single predominant DoF with CC > 0.8 and this DoF did not appear coordinated with other DoF in other groups, and was therefore selected as the independent DoF that has to be measured (Table 2): CMC1A, CMC1F, MCP1F, IP1F, PIP2F and palmar arch. Although groups 10 and 11 (MCP3–4A and MCP2–3A) showed a predominance in only one DoF, they also appeared coordinated with other DoF in group 13. They were thus considered only as candidates to represent the coordination described afterwards (Table 2). The remaining five groups represented the coordination that exists between DoF (Figure 4): (i) PIP coordination (group 1), (ii) MCP coordination (groups 6, 8 and 12), and (iii) coordination between Palmar arch and PIP2F (group 14). This last coordination was discarded as it was present in only two subjects. The predominant DoF listed in Table 1 for the groups representing the two main coordinations between DoF (Figure 5: groups 1, 6, 8 and 12) were considered as candidates to represent them (Table 2).

### 3.3. Selection of the Best Combination of Angles

For the estimation of the physiological angles, eight representative angles were proposed: the six independent DoF (Thumb IP flexion, MCP flexion and CMC flexion and abduction movements, palmar arch and index PIP flexion) as well as two DoF candidates for representing the coordinated movements (one DoF selected from three candidates, and another DoF selected from three candidates). To know from which pair of candidates DoF obtained better estimation joint angles, all possible combinations of them (21 combinations) were tested.

Table 3 shows RMSE errors between estimated and recorded joint angles, for each combination of DoF tested. MCP abduction/adduction obtained low RMSE errors, regardless of the pair of DoF selected. Specifically, MCP2–3A showed errors in the range of 5.7–7.3, MCP3–4A in the range of 4.5–5.4, and MCP4–5A in the range of 3.2–4.4. The estimation of the PIP movement provided better results when the PIP of the finger adjacent to the estimated one was used as representative (e.g., the PIP joint of the middle finger obtained the lowest errors when the PIP of the ring finger was used as representative). The flexion/extension of the MCP joints were the DoF most affected by the selected candidate angles, with a range (maximum-minimum) value of 8 degrees in the case of MCP5F. Similar to PIP joints, the lowest errors were found when the flexion/extension of the MCP joint of the finger adjacent to the estimate was selected. In general, MCP4–5A presented the lowest error and MCP5F presented the highest.

Figure 6 shows the RMSE averaged across DoF. In general, the best estimations were obtained in cases # 10 and # 11. These cases correspond to the usage of the PIP of the ring finger and the MCP of the middle or ring finger as representative.

### 3.4. Evaluation of the Goodness of the Method

To evaluate the goodness of the method, the case with the lowest RMSE was selected (case 11). Accordingly, the representative joint angles finally selected were: CMC1A, CMC1F, MCP1F, IPF1, PIP2F, PalmArch, PIP4F, and MCP4F. Table 4 shows the coefficients x(k) used to estimate all the other joint angles, in accordance with Equation (1). Note that some of the representative angles, such as IP1F and PalmArch, were barely used to estimate the remaining angles, since the coefficients obtained were less than 0.1. MCP4–5A showed low coefficients for all the representative angles, with the highest value for MCP4F movement (−0.13).

Table 5 presents the global RMSE (computed from all frames, subjects, and activities) obtained from the representative angles selected. RMSE values are similar to those of experiment A (Table 3, case 11).

For a better comparison, Figure 7 shows the global RMSE errors expressed as a percentage with respect to the RoM of each DoF considering the angles for both experiment A and experiment B. All the estimated angles presented similar RMSE values, with RMSE errors below 10% of their maximum RoM. Note that the DoF with the highest error were those of the index finger. The lowest error was for flexion/extension of the PIP of the middle finger.

Figure 8 shows the global RMSE errors, expressed as a percentage with respect to the RoM of each DoF, per action and joint (computed from all frames and subjects of each action) for both Experiment A (A1–A26) and Experiment B (B1–B33). All the actions presented RMSE errors, on average, below 10%. For actions from Experiment A, those with the highest errors were “A11. Unscrewing two lids and leaving them on a table” and “A20. Picking up the phone” (AVG = 9.2%). The one with the lowest error was “A5. Lifting and moving an iron” (AVG = 5.0%). For actions from Experiment B, the one with the highest error was “B33. Cooking, serving and cutting a big omelette” (AVG = 9.0%) and the one with the lowest error was “B21. Pouring baking powder into the bowl” (AVG = 6.1%). 

As an example, Figure 9 and Figure 10 show the temporal evolution of the estimated angle versus the recorded angle for subject 2 during two actions from Experiment B: the one with the highest RMSE on average (Figure 9) and the one with the lowest error (Figure 10).

## 4. Discussion

In this work, we identified the minimum set of hand DoF that best represent the hand kinematics, in order to reduce the number of DoF needed to record the whole hand kinematics without losing relevant information by using kinematic synergies. We have provided a detailed description of the most independent hand DoF as well as the most coordinated ones during the performance of representative ADL. Furthermore, we provide the information on all the estimation errors, so that, depending on the field of application, the designer can decide which hand DoF to measure and, therefore, the level of accuracy required. The study was performed according to standardized actions based on the Sollerman Hand Function Test, these tasks reflecting an accurate representativeness of hand functions during ADL.

### 4.1. Hand Kinematic Synergies

The number of synergies per subject required to explain more than 95% of the total variance of the dataset is at least twice the number of synergies extracted in other works: 10 vs. 2–5 [31,32,36,37,45]. This fact may be related to the high percentage of variance explained (95% versus 80–85%) as well as the data standardization procedure followed, which allowed us to compare joints with different RoM. The subsequent cluster analysis grouped the 220 PCs into 14 groups from which we reduced the dataset size (16 DoF) to a limited set of DoF (8 DoF): 6 DoF were obtained from six grouped PCs that represented predominant movement of only 1 DoF (Thumb flexion/abduction of CMC, and flexion of thumb MCP and IP joints, index PIP joint and palmar arch) and the other two DoF were obtained from two coordinated movements (PIP flexion coordination and MCP flexion/abduction coordination). Groups 6, 8 and 12, despite being different, all represent coordination between MCP joints, albeit with different weights, which could correspond to anatomical differences across subjects, among other things. These two coordinated movements obtained herein were similar to those reported in the literature [31,32,36,41,43,48]. However, PIP coordination showed slight differences: this coordinated movement revealed flexion of most fingers except the index, which is consistent with recent studies describing the independence of the index finger and of the thumb [43,49,50]. In this sense, this fact is in accordance with the appearance of all thumb joints and PIP index joint as independent DoF. Palmar arch was not considered in most previous studies because the PCA methodology followed in those studies neglected DoF with small RoM. In previous studies that did take Palmar arch into account [31,43], different coordination between Palmar arch and other joints was found, depending on the activities performed. In this study, where higher accuracy is pursued (by means of the high level of variance sought), a coordination was found between palmar arch and PIP flexion of the index finger, but only in two subjects. These results differ from those reported in previous work [40,41] that used kinematic synergies to recognize eight hand grasping poses from five hand joint movements (thumb abduction of CMC, flexion of MCP of the middle finger, flexion and abduction of MCP of the little finger, and flexion of the PIP of the ring finger). They showed good performance recognition, which demonstrated the value of using kinematic synergies to reduce the complexity of the hand kinematics by recording only a few joint angles. However, several limitations need to be mentioned. First, the method they used undervalued DoF with small RoM: Synergies in PCA using the covariance matrix are quite sensitive to the variances of the original variables so that those with larger ranges of variation dominate over those with small ranges. In order to ensure each variable contributes equally to the analysis, PCA may be applied to the covariance matrix with standardized data (mean = 0 and SD = 1 for each DoF) [44], as has been carried out in this study. This results in a higher number of synergies identified per subject. Second, they considered synergies extracted from only one subject, and were therefore not representative of the overall population. As we have seen, all subjects share the same first two synergies (PIP flexion coordination and MCP flexion/abduction coordination) while the rest of the synergies seem to depend on the different strategies employed by each specific subject [43,44]. Third, the synergies were extracted from non-representative activities (free motion while imagining grasping 57 imaginary objects). However, previous studies have shown differences in coordination between free motion and grasping real objects, and shown that higher PCs differ depending on the tasks or grasps considered [31]. All these facts are expected to lead to poor precision when recording the whole hand kinematics during daily life activities using only the five joint angles proposed by Ciotti and colleagues [40,41]. To make up for those shortcomings, in this work the synergies have been extracted from a representative group of subjects and activities, the result being subject-specific synergies obtained through a method that involves all joint movements equally and makes it possible to explain more than 95% of the variability during ADL.

### 4.2. Estimation of Physiological Angles

Estimation of physiological angles by using the representative DoF selected resulted in low RMSE values for all the combinations tested (Table 3), with maximum average value of 14.2 degrees for the MCP flexion/extension of the little finger and minimum average value of 4.1 degrees for the MCP abduction/adduction of ring and little fingers. In general, MCP and PIP joints, during the flexion/extension movements, obtained lower errors in the estimated angles when using the MCP or PIP flexion/extension movement of adjacent fingers (e.g., MCP3 obtained better results when MCP2 or MCP4 were used as the representative angle). In these cases, the RMSE values obtained were about 9 degrees, in the case of MCP joints, and about 7–10 degrees in the case of PIP joints. In MCP abduction/adduction movements, different behavior was observed, depending on each finger:Abduction/Adduction between index and middle fingers obtained the lowest errors when using MCP flexion/extension movement of the index finger as representative.Abduction/Adduction between middle and ring fingers obtained the lowest errors when using Abduction/Adduction between index and middle fingers as representative.Abduction/Adduction between ring and little fingers obtained the lowest errors when using MCP flexion/extension movement of the little finger as representative.

In particular, for each of the 21 combinations tested (Table 3), the best solution, in terms of lower average RMSE across DoF, was the combination that used flexion of MCP and PIP joints of the ring finger (Figure 11). This result seems quite reasonable, since the thumb and index finger are more independent, and the one that best approaches the others is the ring, which is more centered with the other two fingers. In this case, PIP index finger flexion was the DoF with the highest error (12.14 degrees, 9.8% with respect to its RoM) and relative abduction MCP4–5 was the DoF with the lowest error (3.89 degrees, 6% with respect to its RoM). These specifications could be helpful in designing simpler devices to record the whole hand kinematics, by reducing the number of DoF to be recorded, but obtaining the best possible estimation angles for the non-recorded DoF. In fields such as virtual reality, recording eight DoF (the best solution found herein) would be sufficient to obtain the complete kinematics of the hand, considering that these estimations were made from common grasps and objects used in everyday life. In other fields, like teleoperation, or in other applications where a higher precision in specific DoF is necessary, other combinations of DoF might be recorded. For this purpose, Table 3 presents all the combinations tested that could be used to select other combinations of DoF, depending on which joint and finger require more accuracy. As an example, if more precision is needed in the index and middle fingers, MCP2F and PIP3F may improve estimation angles for these fingers. In the case of needing more precise movements, or other activities that do not require grasping (such as pointing with a finger), the usefulness of these simplifications should be studied in more detail.

The proposed solution presents different coefficients for each representative DoF selected, depending on the DoF to be estimated (Table 4). From these coefficients, we can observe which representative DoF are used to estimate each remaining DoF: MCP flexion/extension movement of the index and middle fingers are mainly estimated from flexion/extension of the MCP joints and CMC thumb joint. This means that MCP movements of these fingers are highly related to thumb position and MCP movement of the other fingers.In general, PIP joints are mainly related to the PIP joint of the adjacent finger. In particular, PIP of the middle finger presents a mainly high relation to PIP of the ring finger, with some influence from PIP of the index finger and CMC thumb flexion. PIP of the little finger mainly presents a relation with PIP of the ring finger with some influence from MCP of the ring finger.MCP abduction/adduction movement between index, middle and ring fingers are more related to thumb CMC flexion/extension and MCP flexion/extension of ring finger. This means that abduction/adduction movement is also related to the thumb position (in this case only with CMC flexion movement) and MCP flexion/extension movements. MCP abduction/adduction between the ring and little fingers are only related to MCP flexion/extension movement.

### 4.3. Evaluation of the Goodness of the Method 

The method presented herein obtained similar errors in different datasets regardless of the type of complexity of the ADL to be performed. In experiment A, the ADL were more controlled and repeatable between subjects whereas in experiment B subjects had more freedom to carry out the activities, which were more complex and varied, since each of them involved several actions. In general, average errors obtained were below 10% of the RoM of each DoF, for all the activities and DoF. The estimations were performed with the hand movements of a few representative daily life activities (Experiment A) and were then applied to continuous and complex actions (Experiment B), involving reaching, grasping, and manipulation stages, with varied products and actions (beating, stirring, cooking, serving, cutting, etc.), as well as natural and free hand movements during those actions. 

The estimates presented bounded and limited errors, with values between 3.1% and 16.8% of the RoM of each DoF, depending on the DoF estimated and the activity performed. From Experiment A, activities A11 and A20 (“A11. Unscrewing two lids and leaving them on a table” and “A20. Picking up the phone”) presented the highest estimation errors, with both activities using the intermediate power-precision grasp predominantly. From experiment B, activity B33 (“Cooking, serving and cutting a big omelet”) presented the highest estimation error. This activity consists in: (1) Taking the lid of the pan from the worktop, putting it over the pan and turning the omelet over with the lid; (2) Grasping the handle of the pan and shaking it; (3) Taking the pan by the handle and putting the omelet on the plate; (4) Taking the knife from the worktop and cutting the omelet into four pieces and leaving the knife on the worktop. Figure 8 shows that the highest estimation error comes from the first part of the activity, which corresponds to the action of grasping the handle of the pan (i.e., during intermediate power-precision grasp). This means that the highest errors occurred when the object to be handled required the performance of an intermediate power-precision grasp. This fact makes it more difficult to estimate the position, in this case, of the MCP of the index finger (error of 16.8%), using the rest of the DoF. Flexion of MCP of the index finger was estimated mainly from flexion of ring MCP and thumb CMC movements. Therefore, the highest estimation error was produced when intermediate power-precision grasp was performed, producing a different relationship between these joint movements, as in the case of activities A11, A20 and B33. However, this grasp type is barely used during the performance of ADL (3.3%) and personal autonomy (7.9%), according to previous studies [42,51].

In contrast, from Experiment A, activity A5 (“Lifting and moving an iron”) presented the lowest error and was characterized by a cylindrical grasp. From Experiment B, the lowest error was observed in activity B21 (“Pouring baking power into the bowl”), which consisted only in: opening the box of baking powder, taking out one sachet and opening it, pouring the powder into the bowl, and closing the box. This activity was mainly characterized by using a pad-to-pad pinch. Cylindrical grasp and pad-to-pad pinch are the most used during ADL and personal autonomy (38.3% and 12.6% during ADL, and 33.4% and 15.3% for personal autonomy, respectively) [42,51]. Therefore, by using the eight DoF selected, a high level of estimation is obtained during the most common grasps used during ADL and personal autonomy.

## 5. Conclusions

This paper proposes a reduced set of hand DoF that best represent the hand kinematics, in order to reduce the number of joint angles needed to obtain the whole hand kinematics through the use of kinematic synergies. The results could be helpful in the design of simpler devices to record the whole hand kinematics, by reducing the number of DoF to be recorded, but obtaining the best possible estimation angles for the non-recorded DoF. In fields such as virtual reality, recording eight joint angles (the best solution found in this study) would be sufficient to obtain the complete kinematics of the hand, considering that these estimations were made from common grasps and objects used in everyday life. In other fields like teleoperation, or in other applications where a higher precision in specific DoF is necessary, other combinations of joint angles could be recorded. This reduced set of hand joints could diminish some problems such as those of occlusion, placement times and post-follow-up processing times, as well as the investment required. Note that the sample used herein may not be representative of possible kinematic variabilities that may appear in some specific pathological conditions. Therefore, the proposal presented herein may not be suitable in those cases, for which specific studies would be needed.

## Figures and Tables

**Figure 1 sensors-21-01049-f001:**
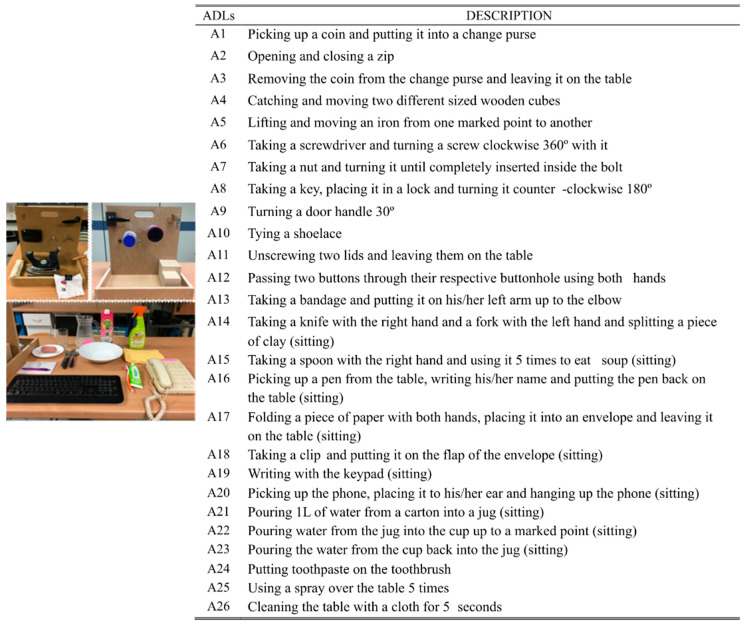
Experiment A: List of activities of daily living (ADLs) performed and scenario with the objects used. Unless indicated otherwise, the position of the subject was standing.

**Figure 2 sensors-21-01049-f002:**
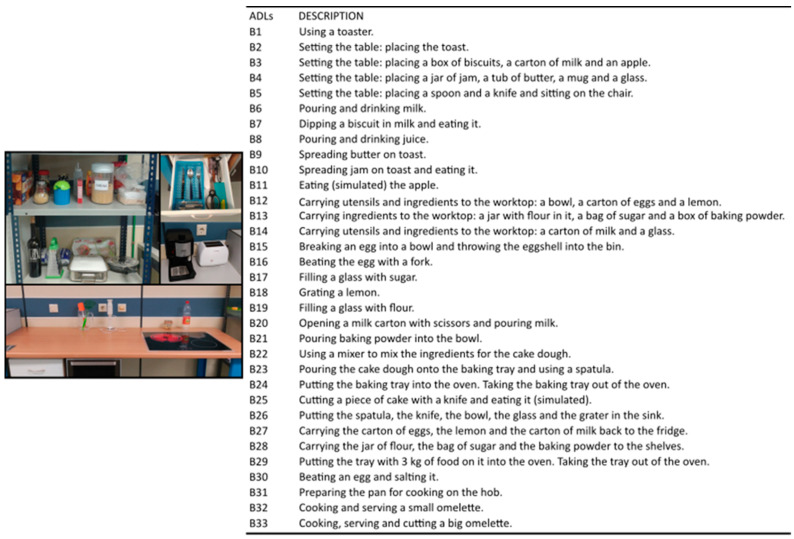
Experiment B: List of ADLs performed and scenarios with the objects used to evaluate the goodness of the method.

**Figure 3 sensors-21-01049-f003:**
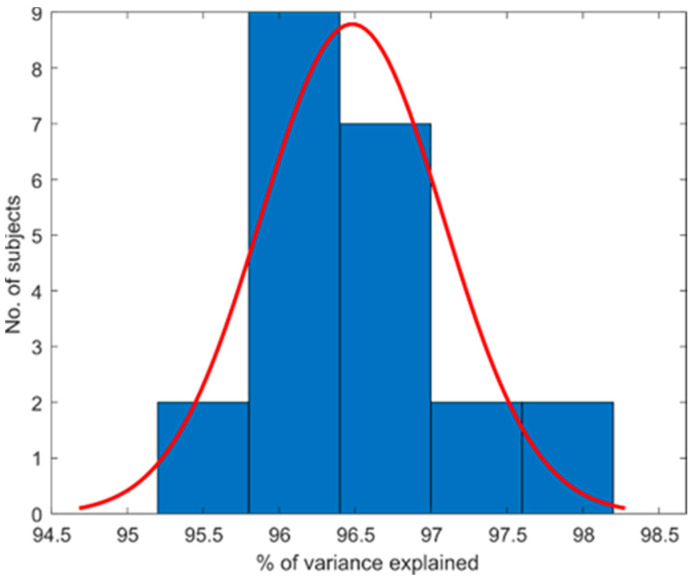
Total amount of variance explained per subject.

**Figure 4 sensors-21-01049-f004:**
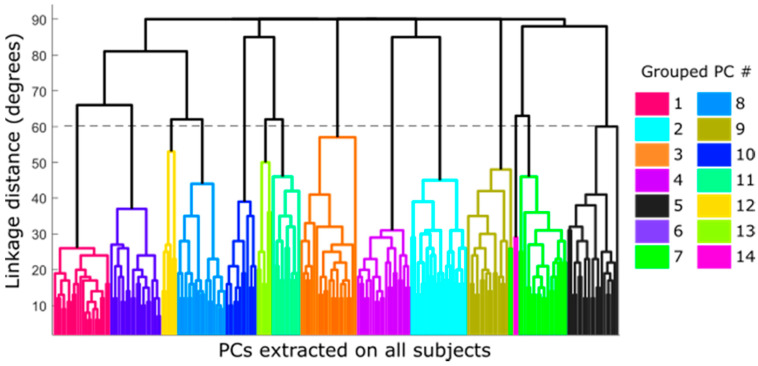
Dendrogram for cluster composition of the extracted principal components (PCs). The horizontal axis represents all the PCs with 220 numbers of clustered nodes; vertical axis represents the pairwise distance between the observed PCs.

**Figure 5 sensors-21-01049-f005:**
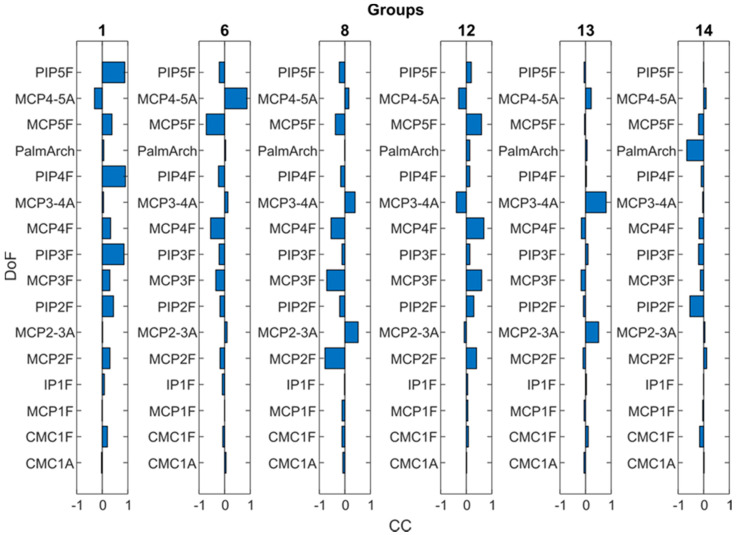
Weight of each DoF movement of the 6 grouped PCs that represent coordination between DoF. 1 to 5 refers to thumb to little fingers. F stands for flexion/extension and A for abduction/adduction movements.

**Figure 6 sensors-21-01049-f006:**
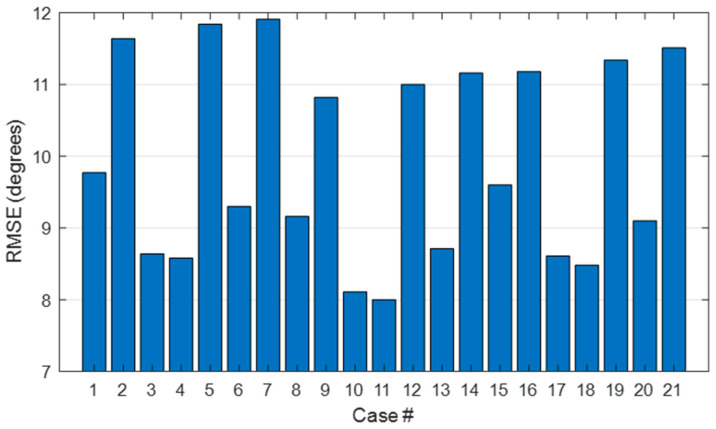
Averaged RMSE across DoF obtained for each case number tested.

**Figure 7 sensors-21-01049-f007:**
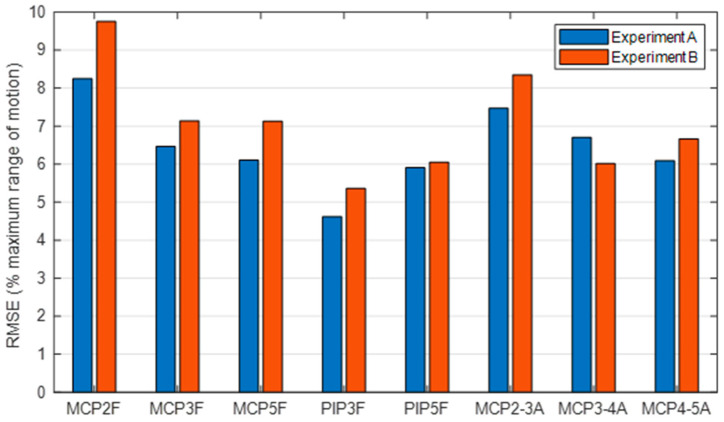
Global RMSE per joint (case #11) computed from all the frames and expressed as a percentage with respect to the RoM of each DoF for both experiments A and B. 1 to 5 refers to thumb to little fingers. F stands for flexion/extension and A for abduction/adduction movements.

**Figure 8 sensors-21-01049-f008:**
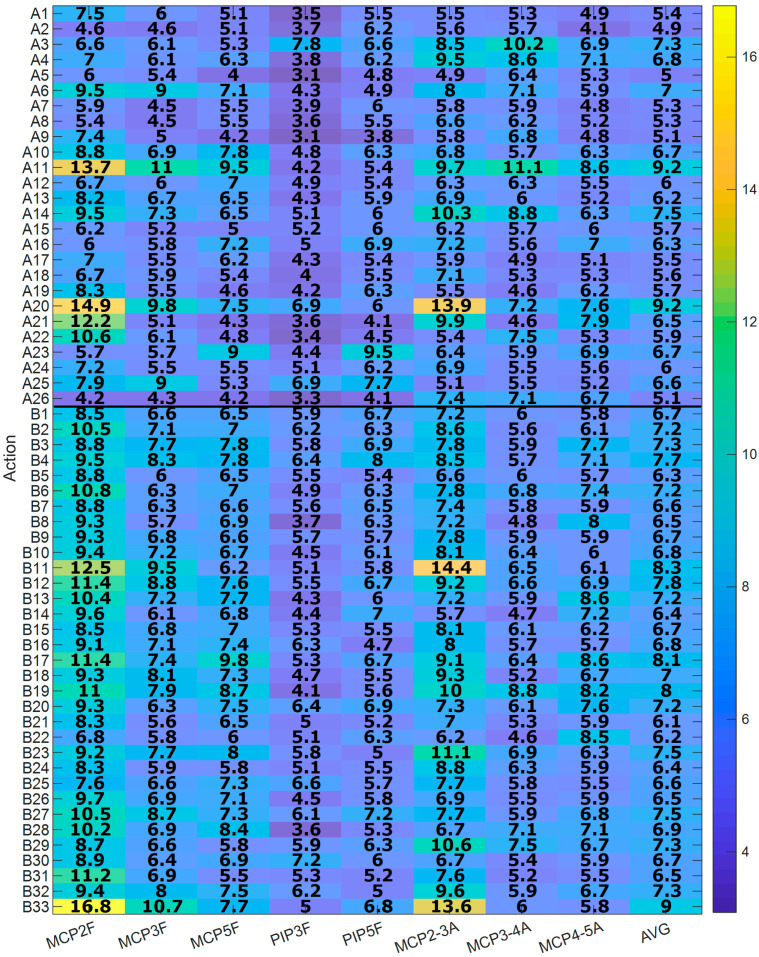
RMSE per joint and action (Experiment B), expressed as a percentage with respect to the RoM of each DoF. The last column represents the average between DoF (AVG). 1 to 5 refers to thumb to little fingers. F stands for flexion/extension and A for abduction/adduction movements.

**Figure 9 sensors-21-01049-f009:**
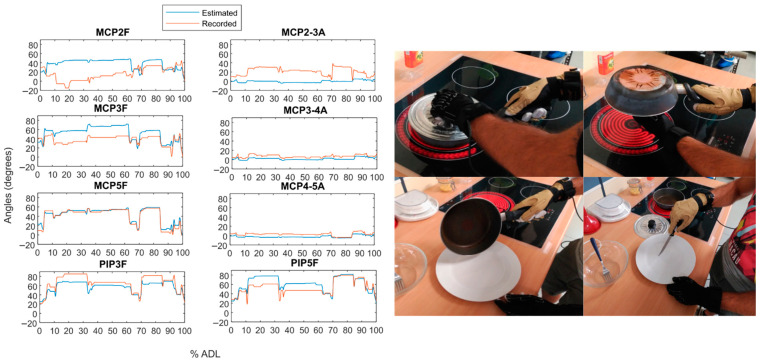
Estimated versus recorded angles for action 33 (“Cooking, serving and cutting a big omelette”). 1 to 5 refers to thumb to little fingers. F stands for flexion/extension and A for abduction/adduction movements.

**Figure 10 sensors-21-01049-f010:**
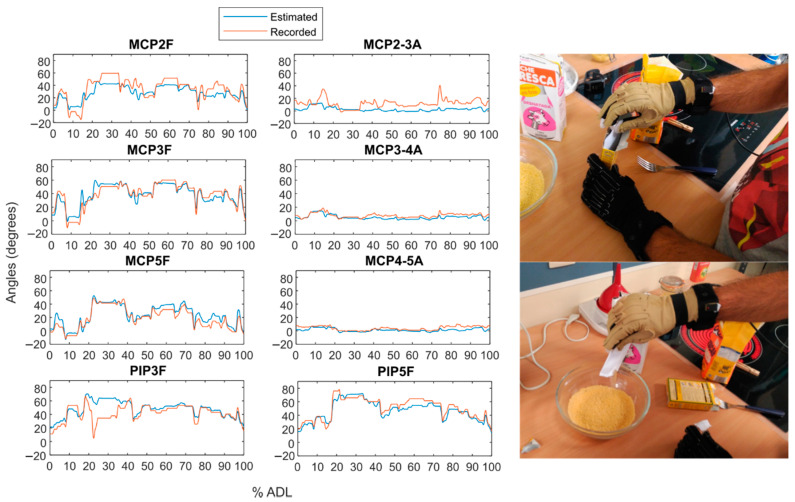
Estimated versus recorded angles for action 21 (“Pouring baking powder into the bowl”). 1 to 5 refers to thumb to little fingers. F stands for flexion/extension and A for abduction/adduction movements.

**Figure 11 sensors-21-01049-f011:**
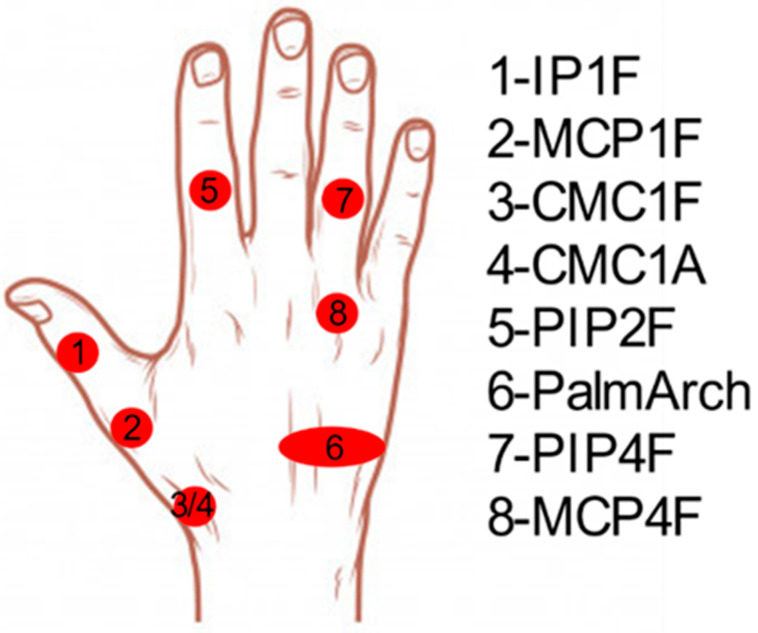
Best 8 DoF selected as representative angles for the estimation of the remaining hand joints. 1 to 5 refers to thumb to little fingers. F stands for flexion/extension and A for abduction/adduction movements.

**Table 1 sensors-21-01049-t001:** Percentage of subjects, predominant degrees of freedom (DoF) (sorted from higher to lower correlation coefficients) and description of each group of synergies. F for flexion/extension and A for abduction/adduction movements. 1 to 5 for thumb to little digits.

Groups	Subjects (%)	Predominant DoF	Description
1	100	PIP4F, PIP5F PIP3F PIP2F	Fingers 2–5 PIPF coordination
2	100	CMC1A	Thumb CMCA movement
3	100	MCP1F	Thumb MCPF movement
4	95.5	IP1F	Thumb IPF movement
5	90.9	CMC1F	Thumb CMCF movement
6	90.9	MCP4–5A, MCP4F, MCP5F	Fingers 2–5 MCP coordination (more weight from Fingers 4 and 5)
7	86.4	PIP2F	Index PIPF movement
8	86.4	MCP3F, MCP2F, MCP4F	Fingers 2–5 MCP coordination (more weight from Fingers 2–4)
9	81.8	PalmArch	Palmar arch movement
10	54.5	MCP2–3A	MCP2–3A movement
11	50.5	MCP3–4A	MCP3–4A movement
12	27.2	MCP2F, MCPF2–3A, MCP3F, MCP4F, MCP3–4A	Fingers 2–5 MCP coordination (with any DoF predominance)
13	27.2	MCP3–4A, MCP2–3A	MCP2–3A and MCP3–4A coordination
14	9.0	PalmArch, PIP2F	PalmArch, and PIP2F coordination

**Table 2 sensors-21-01049-t002:** Independent and candidate DoF for the estimation of physiological angles. F for flexion/extension and A for abduction/adduction movements. 1 to 5 for thumb to little digits.

Independent DoF	DoF Candidate of PIP Coordination (from Group 1)	DoF Candidate of MCP Coordination (from Groups 6, 8 and 12)
CMC1A	PIP3F	MCP2F
CMC1F	PIP4F	MCP2–3A
MCP1F	PIP5F	MCP3F
IPF1		MCP4F
PIP2F		MCP3–4A
PalmArch		MCP5F
		MCP4–5A

**Table 3 sensors-21-01049-t003:** RMSE between estimated and recorded joint angles across subjects and actions (for each combination of candidate DoF). 1 to 5 refers to thumb to little fingers. F stands for flexion/extension and A for abduction/adduction movements.

	Candidate DoF		RMSE (Degrees)									
Case	1	2	MCP2F	MCP2–3A	MCP3F	PIP3F	MCP4F	MCP3–4A	PIP4F	MCP5F	MCP4–5A	PIP5F
1	PIP3F	MCP2F		5.7	10.5		13.7	4.8	9.2	16.0	4.3	13.9
2	PIP3F	MCP2–3A	12.0		15.4		15.9	4.5	9.4	17.3	4.4	14.1
3	PIP3F	MCP3F	9.0	6.3			9.1	4.6	9.1	13.3	4.2	13.6
4	PIP3F	MCP4F	12.2	6.7	9.4			4.6	9.3	8.7	3.9	13.8
5	PIP3F	MCP3–4A	13.6	6.1	15.2		14.6		9.5	17.2	4.3	14.2
6	PIP3F	MCP5F	13.8	7.1	13.3		8.4	5.2	9.3		3.2	14.0
7	PIP3F	MCP4–5A	14.8	7.2	16.8		14.9	5.3	9.5	12.6		14.2
8	PIP4F	MCP2F		5.8	10.5	7.6	13.7	4.9		16.0	4.3	10.4
9	PIP4F	MCP2–3A	11.8		15.0	7.5	15.7	4.6		17.1	4.4	10.4
10	PIP4F	MCP3F	9.0	6.3		7.5	9.3	4.6		13.6	4.2	10.3
11	PIP4F	MCP4F	12.1	6.8	9.7	7.7		4.7		8.8	3.9	10.3
12	PIP4F	MCP3–4A	13.3	6.1	14.8	7.6	14.4			16.9	4.3	10.5
13	PIP4F	MCP5F	13.7	7.2	13.6	7.7	8.5	5.3			3.2	10.4
14	PIP4F	MCP4–5A	14.6	7.2	16.7	7.7	14.8	5.3		12.5		10.5
15	PIP5F	MCP2F		5.9	10.5	11.2	13.6	4.9	10.2	16.2	4.3	
16	PIP5F	MCP2–3A	11.8		14.8	10.9	15.5	4.6	10.1	17.3	4.4	
17	PIP5F	MCP3F	9.0	6.3		11.1	9.3	4.6	10.2	13.9	4.2	
18	PIP5F	MCP4F	12.1	6.9	9.7	11.3		4.7	10.2	9.1	3.9	
19	PIP5F	MCP3–4A	13.2	6.1	14.5	11.1	14.2		10.1	17.1	4.4	
20	PIP5F	MCP5F	13.6	7.2	13.6	11.1	8.5	5.3	10.1		3.2	
21	PIP5F	MCP4–5A	14.5	7.3	16.5	11.2	14.6	5.4	10.3	12.6		
Statistics across cases	Minimum		9.0	5.7	9.43	7.5	8.4	4.5	9.1	8.7	3.2	10.3
	Maximum		14.8	7.3	16.8	11.3	15.9	5.4	10.2	17.3	4.4	14.2
	Average		12.5	6.6	13.4	9.4	12.7	4.9	9.7	14.2	4.1	12.2

**Table 4 sensors-21-01049-t004:** Coefficients x(k) obtained for each of the representative joint angles, in order to estimate Table 1. All the coefficients presented are significant with a *p*-value of 0.05. 1 to 5 refers to thumb to little fingers. F stands for flexion/extension and A for abduction/adduction movements.

	Coefficients x(k)								
Estimated Angles	Intercept	CMC1F	CMC1A	MCP1F	IP1F	PIP2F	PIP4F	MCP4F	P_Arch
MCP2F	8.66	-	−0.24	0.32	−0.01	−0.04	0.06	0.52	0.01
MCP2–3A	3.53	−0.05	−0.19	−0.04	0.01	−0.03	0.06	−0.16	0.02
MCP3F	8.91	-	−0.21	0.19	−0.07	0.08	−0.04	0.87	0.08
PIP3F	−0.05	−0.05	−0.07	−0.03	−0.02	0.19	0.69	−0.02	0.02
MCP3–4A	3.67	−0.07	−0.15	−0.01	0.01	−0.01	0.06	−0.17	0.05
MCP5F	−5.78	−0.14	−0.01	−0.07	0.05	−0.03	0.15	0.89	−0.04
MCP4–5A	5.49	0.00	−0.04	0.05	−0.03	0.00	−0.03	−0.13	−0.01
PIP5F	1.89	0.07	−0.04	0.06	0.05	−0.06	0.82	0.13	0.01

**Table 5 sensors-21-01049-t005:** Root mean square errors (RMSE) per joint obtained for Experiment B computed from all frames. 1 to 5 refers to thumb to little fingers. F stands for flexion/extension and A for abduction/adduction movements.

RMSE (Degrees)							
MCP2F	MCP2–3A	MCP3F	PIP3F	MCP3–4A	MCP5F	MCP4–5A	PIP5F
14.35	7.63	10.68	8.92	4.18	10.25	4.25	10.54

## Data Availability

The data presented in this study are openly available in Zenodo at https://doi.org/10.5281/ZENODO.3469380 and in Mendeley Data at https://doi.org/10.17632/8mf4y2srgh.

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
