# Peer review of "Synergy-Based Sensor Reduction for Recording the Whole Hand Kinematics"

_sensors, 2021, doi:10.3390/s21041049_

Round 1

Reviewer 1 Report

The authors present a method of reducing the number of sensors on a hand by analyzing the synergistic movement of the hand in daily activities of living. They have succeeded in halving the number of sensors needed to record the entire hand kinematics. The presentation is novel and interesting and suitable for publication.

Herein, I recommend only 1 suggestion for the authors:

1) It is not known the ages and genders of the subjects in the experiments. Would there be a much bigger variation in elderly subjects, who would actually benefit the most from this study towards understanding their ADLs?

Author Response

Reviewer #1 (Remarks to the Authors):

 The authors present a method of reducing the number of sensors on a hand by analyzing the synergistic movement of the hand in daily activities of living. They have succeeded in halving the number of sensors needed to record the entire hand kinematics. The presentation is novel and interesting and suitable for publication.

 Herein, I recommend only 1 suggestion for the authors:

 1) It is not known the ages and genders of the subjects in the experiments. Would there be a much bigger variation in elderly subjects, who would actually benefit the most from this study towards understanding their ADLs?

RESPONSE: Many thanks to the reviewer for the suggestion. We fully agree with the reviewer, the sample used herein may not be representative of possible variabilities that may appear in some specific pathological conditions, which would limit its use in that population. We have added subjects’ characteristics along with inclusion and exclusion criteria in Material and Methods section, and we have added a limitation in page 18, lines 585-587: “Note that the sample used herein may not be representative of possible kinematic variabilities that may appear in some specific pathological conditions. Therefore, the proposal presented herein may not be suitable in those cases, for which specific studies would be needed”.

Reviewer 2 Report

In this study, the authors analyzed a synergy-based reduction method for the analysis of the hand kinematic synergy. Despite the paper is interesting, some major revision should be considered.

Page 1 lines 38-39 “Electrogoniometers are commonly used in clinical practice to measure the range of motion (RoM) of joints.” Please add references to this sentence.

Page 2 lines 48-50 “Methods with one or two markers are prone to larger skin movements, because the joint heads of the fingers have many wrinkles in 49the skin.” Please add a reference to this sentence.

Page 2 lines 54-55 “These systems are easy to transport and manipulate and measure  captured kinematic  information  with high  accuracy.” A reference demonstrating the high accuracy of the kinematic measure through IMUs in the analysis of hand synergies should be added.

Page 2 lines 56-57 “However, they are affected by movements of the skin and possible disturbances of the magnetic field [11] and their size limits their use for measuring hand kinematics”. Inertial sensors present other limitations such as the orientation algorithm (Kalman filter, complementary filter), body to sensors alignment method along with drift, noise, temperature influence. Please consider these limitations.

Page 2 lines 76-77 “Principal component analysis (PCA) is the most used statistical method for obtaining kinematic synergies”. In a recent study (Lambert-Shirzad 2017), a comparison between factorization methods for the extraction of kinematic synergies highlighted common performances between PCA and NNMF. As the author suggested, kinematic synergies extracted via NNMF method resulted in more physiologically meaningful. In addition, the NNMF method was found to be the most appropriate method also in the extraction of muscle synergies as attested by a recent study of Rabbi and colleagues in 2020. The authors should consider these literature findings and clarified the choice of using PCA instead of NNMF.

Lambert-Shirzad, N., & Van der Loos, H. M. (2017). On identifying kinematic and muscle synergies: a comparison of matrix factorization methods using experimental data from the healthy population. Journal of neurophysiology, 117(1), 290-302.

Rabbi, M. F., Pizzolato, C., Lloyd, D. G., Carty, C. P., Devaprakash, D., & Diamond, L. E. (2020). non-negative matrix factorisation is the most appropriate method for extraction of muscle synergies in walking and running. Scientific Reports, 10(1), 1-11.

Page 2 lines 77-79 “Previous studies have shown that a few linear combinations of the hand joint movements (principal components, PCs) could account for most of the variance in the original set of hand postures.” Please add references to this sentence.

Material and methods section

As the authors stated, kinematic data of twenty-two right-handed subjects were used in the analysis. To improve the consistency of the analysis subjects’ characteristics along with inclusion and exclusion criteria should be added.

Description of the sensors/instrumentation used to gathered kinematic data should be added and exhaustively reported.

A more in-depth description of the protocol should be reported in the text to make the reading and interpretation of the analysis clearer. How were the motor tasks performed? Were the motor tasks randomized across subjects? How many repetitions did each subject for each task perform?

Page 4 lines 154-156 Why all the obtained angles were filtered with a low-pass filter at 5 Hz? Is this approach shown in some other work in literature?

Page 4 line 161 Why all the angle time profiles were 1000 frames long? How does the angle signals were normalized? Please specify.

Page 5 lines-190-194: “CC > 0.8 for 194only one DoF, and CC < 0.3 for all other DoF” and “CC > 0.4”. Why the authors have chosen these thresholds? Is this approach reported in some other findings? 

Page 6 lines 227-228. Similar to the first database, the subjects’ characteristics of the sample and a deeper description of the experimental protocol should be reported.

Result

Page 11. Please report the standard deviation of the RMSE among subjects and activities in table 3, table 5, figure 7, and among subjects in figure 8.

Page 11 lines 350 352 “Note that the DoF with the highest error were those of the index finger. The lowest error was for flexion/extension of the PIP of the middle finger.” Since only average values of RMSE are considered, these considerations are just qualitative. An ANOVA test should be considered for the evaluation of the highest and the lowest errors

Page 12. Figure 8. Similar consideration of the previous comment. A statistical analysis of the results of the RMSE should be added for the assessment of the lower error.

To improve the readability of the results, a full description of the acronyms should be added in all the capitation of both figures and tables.

Author Response

Reviewer #2 (Remarks to the Authors):

 In this study, the authors analyzed a synergy-based reduction method for the analysis of the hand kinematic synergy. Despite the paper is interesting, some major revision should be considered.

 RESPONSE: Many thanks to the reviewer for his/her thorough revision and all suggestions made. We believe that after all changes made, the new version of the paper has improved in quality and clarity. The changes have been highlighted in yellow.

Page 1 lines 38-39 “Electrogoniometers are commonly used in clinical practice to measure the range of motion (RoM) of joints.” Please add references to this sentence.

RESPONSE: We have added a reference that supports this sentence (page 2, line 56).

Page 2 lines 48-50 “Methods with one or two markers are prone to larger skin movements, because the joint heads of the fingers have many wrinkles in 49the skin.” Please add a reference to this sentence.

 RESPONSE: We have added references that support this sentence (page 2, line 66).

Page 2 lines 54-55 “These systems are easy to transport and manipulate and measure captured kinematic information with high accuracy.” A reference demonstrating the high accuracy of the kinematic measure through IMUs in the analysis of hand synergies should be added.

RESPONSE: We have added a reference that supports this sentence (page 2, line 72).

Page 2 lines 56-57 “However, they are affected by movements of the skin and possible disturbances of the magnetic field [11] and their size limits their use for measuring hand kinematics”. Inertial sensors present other limitations such as the orientation algorithm (Kalman filter, complementary filter), body to sensors alignment method along with drift, noise, temperature influence. Please consider these limitations.

 RESPONSE: We agree. All those other limitations have been added (page 2, lines 72-76).

Page 2 lines 76-77 “Principal component analysis (PCA) is the most used statistical method for obtaining kinematic synergies”. In a recent study (Lambert-Shirzad 2017), a comparison between factorization methods for the extraction of kinematic synergies highlighted common performances between PCA and NNMF. As the author suggested, kinematic synergies extracted via NNMF method resulted in more physiologically meaningful. In addition, the NNMF method was found to be the most appropriate method also in the extraction of muscle synergies as attested by a recent study of Rabbi and colleagues in 2020. The authors should consider these literature findings and clarified the choice of using PCA instead of NNMF.

 Lambert-Shirzad, N., & Van der Loos, H. M. (2017). On identifying kinematic and muscle synergies: a comparison of matrix factorization methods using experimental data from the healthy population. Journal of neurophysiology, 117(1), 290-302.

 Rabbi, M. F., Pizzolato, C., Lloyd, D. G., Carty, C. P., Devaprakash, D., & Diamond, L. E. (2020). non-negative matrix factorisation is the most appropriate method for extraction of muscle synergies in walking and running. Scientific Reports, 10(1), 1-11.

RESPONSE: We do not fully agree with the reviewer. Non-negative matrix factorization requires the positiveness of the signals. For kinematics, this implies weird reference postures to ensure that all joint angles are positive. In addition, non-orthogonal rotation does not preserve the independence of the factors and the interpretation of the factorial space becomes less obvious. A Recent work (Prevete et al. 2018) suggests that synergies used by the subjects are intrinsically sparse both in DoF and in actions (i.e., each synergy uses a limited set of DoF and each action is implemented with a combination of a limited number of synergies) and that there is a large set of synergies, shared across subjects (as we have seen in our paper).

PCA with Varimax rotation (as we have applied herein) has been applied before to obtain sparse synergies (Gracia-Ibañez et al., 2020), so that each synergy obtained comprises just a limited number of DoF with very high loadings on this synergy, while the others remain with almost zero loadings. When performing PCA, first factor of the unrotated solutions tends to be a general factor with almost every variable loading significantly, and accounting for a large amount of variance and the subsequent factors are then based on a residual amount of variance. Varimax rotation redistributes the variance from earlier factors to later ones, by maximizing the sum of the variances of the squared loadings, so that all the coefficients will be either large or near zero. Therefore, the use of Varimax rotation overcomes the problem of sparsity in DoF.

The article referred by the reviewer (Lambert-Shirzad et al., 2017) did not consider the varimax rotation and normalization applied here, and therefore the results are not comparable. The second study referred by the author used the NNMF to identify muscle synergies, which are of a different nature from kinematics (activation signal is always positive).

To clarify the reason of using PCA, a sentence has been added (page 2, lines 95-97): “Although there are other methods to compute synergies, PCA is the most used statistical method for obtaining kinematic synergies because it allows sparse synergies to be obtained

Prevete, R., Donnarumma, F., d’Avella, A. & Pezzulo, G. Evidence for sparse synergies in grasping actions. Sci. Rep. 8, 616 (2018).

Gracia-Ibáñez, V.; Sancho-Bru, J.L.; Vergara, M.; Jarque-Bou, N.J.; Roda-Sales, A. Sharing of hand kinematic synergies across subjects in daily living activities. Sci. Rep. 2020, doi:10.1038/s41598-020-63092-7.

Page 2 lines 77-79 “Previous studies have shown that a few linear combinations of the hand joint movements (principal components, PCs) could account for most of the variance in the original set of hand postures.” Please add references to this sentence.

 RESPONSE: We have added references that support this sentence (page 3, line 100).

Material and methods section

 As the authors stated, kinematic data of twenty-two right-handed subjects were used in the analysis. To improve the consistency of the analysis subjects’ characteristics along with inclusion and exclusion criteria should be added.

 RESPONSE: Subjects’ characteristics for both experiments have been added, along with inclusion and exclusion criteria, in order to improve the consistency of the analysis (page 4, lines 156-159, and page 6, lines 257-260). Gender parity was pursued when recruiting participants, who were required to be adults (aged between 20 and 65 years) and free of upper limb pathologies.

Description of the sensors/instrumentation used to gathered kinematic data should be added and exhaustively reported.

RESPONSE: Description of the instrumented glove used to gather kinematic data has been added in the Materials and Methods section (page 5, lines 171-174).

A more in-depth description of the protocol should be reported in the text to make the reading and interpretation of the analysis clearer. How were the motor tasks performed? Were the motor tasks randomized across subjects? How many repetitions did each subject for each task perform?

  RESPONSE: As explained in the manuscript, the detailed protocol is published in a previous work (Jarque-bou et al, 2020). This protocol consisted in performing 26 simulated ADL, 20 of which are included in SHFT. Some ADL from SHFT were adapted to ensure their repeatability, and six further activities (A10, A15, A19, A24, A25, and A26) were added based on the percentage of use of the commonest grasps during ADL. The participants were given clear instructions as to how to perform each task, including details like the angle of rotation of the key (A8), the position of the coin (A1 & A3), the angle of rotation of the door handle (A9) or the amount of water to be poured (A21). Subjects were told to start and end each task in the same posture: arms relaxed on each side of their body, when the subject was standing, or arms resting in a relaxed position on a table when sitting. The subjects could practice each task as many times as necessary in advance to become familiar with its performance before recordings. They performed each activity once, and all subjects did them in the same order.

It was already explained in the manuscript that the detailed protocol can be found in a previous published work (Jarque-bou et al, 2020), but we have added a sentence for a clear reading and interpretation (page 4, line 166): “They performed each activity once, and all subjects did them in the same order”.

Page 4 lines 154-156 Why all the obtained angles were filtered with a low-pass filter at 5 Hz? Is this approach shown in some other work in literature?

 RESPONSE: A low-pass filter is required to avoid measurement noise, as shown by (Schreven et al, 2015), and a 5 Hz cutoff has been used in many works in literature. We have added some references in which this approach has previously used (page 5, line 185).

Sander Schreven, Peter J. Beek, Jeroen B.J. Smeets, Optimising filtering parameters for a 3D motion analysis system, Journal of Electromyography and Kinesiology, Volume 25, Issue 5,2015, Pages 808-814, ISSN 1050-6411, https://doi.org/10.1016/j.jelekin.2015.06.004.

Page 4 line 161 Why all the angle time profiles were 1000 frames long? How does the angle signals were normalized? Please specify.

 RESPONSE: For all the ADL to weigh the same in the analyses, the number of frames of each record were rescaled to 1000 frames per ADL, as previously done in (Jarque-bou et al,. 2020). This information has been added to the manuscript, as well as how the signals were normalised (page 5, lines 191-192 and 195): “the joint angles were normalised by rescaling them to unit variance (mean = 0 and SD = 1 for each DoF) in order to prevent the first PCs from reflecting the joint angles with the largest amplitudes”.

Page 5 lines-190-194: “CC > 0.8 for 194only one DoF, and CC < 0.3 for all other DoF” and “CC > 0.4”. Why the authors have chosen these thresholds? Is this approach reported in some other findings?

 RESPONSE: These values have been selected by following the recommendations of a recent work (Schober et al., 2018), in which the following tentative thresholds are proposed: 0.39 or less for weak correlation, more than 0.7 for strong correlation and more than 0.4 for moderate correlation. We have adapted these thresholds to our field, as suggested by the authors. This reference has been also added in page 5, line 222.

Schober, Patrick MD, PhD, MMedStat; Boer, Christa PhD, MSc; Schwarte, Lothar A. MD, PhD, MBA Correlation Coefficients: Appropriate Use and Interpretation, Anesthesia & Analgesia: May 2018 - Volume 126 - Issue 5 - p 1763-1768 doi: 10.1213/ANE.0000000000002864

Page 6 lines 227-228. Similar to the first database, the subjects’ characteristics of the sample and a deeper description of the experimental protocol should be reported.

 RESPONSE: Subjects’ characteristics have been added also for this experiment, along with inclusion and exclusion criteria (page 4, lines 156-159, and page 6, lines 257-260). Gender parity was pursued when recruiting participants, who were required to be adults (aged between 20 and 65 years) and free of upper limb pathologies.

Result

 Page 11. Please report the standard deviation of the RMSE among subjects and activities in table 3, table 5, figure 7, and among subjects in figure 8.

 RESPONSE: We think that the reviewer did not fully understand the computation of the RMSE. RMSE was computed as global RMSE (using all the frames of all the subjects and activities) or, as in figure 8 (using all the frames for each activity). In any case, there is only one RMSE value per joint for the global RMSE and, in the case of figure 8, one RMSE value per joint and activity. We have now emphasized this fact in the document (page 7, lines 278-279, and page 13, lines 388-389). Therefore, there are not standard deviation values, and for this reason they were not included. In fact, standard deviation and RMSE are similar because they are square roots of squared differences between some values. Nonetheless, they are not the same. Standard deviation is used to measure the spread of data around the mean, while RMSE is used to measure distance between some values and prediction for those values. RMSE is generally used to measure the error of prediction, i.e. how much the predictions made differ from the predicted data, as is used in our paper. In addition, RMSE is a commonly used statistic to show model performance (Weiss and Hays, 2004, Doraiswamy et al., 2005, Lobell et al., 2005). We have rephrased some text in order to clarify this fact.

Weiss, A., Hays, C.J., Hu, Q., Easterling, W.E., 2001. Incorporating bias error in calculating solar irradiance: implications for crop yield simulations. Agron. J. 93, 321–1326

P.C. Doraiswamy, T.R. Sinclair, S. Hollinger, B. Akhmedov, A. Stern, J. Prueger. Application of MODIS derived parameters for regional crop yield assessment. Remote Sens. Environ., 97 (2005), pp. 192-202

D.B. Lobell, J.I. Ortiz-Monasterio, G.P. Asner, R.L. Naylor, W.P. Falcon. Combining field surveys, remote sensing, and regression trees to understand yield variations in an irrigated wheat landscape Agron. J., 97 (2005), pp. 241-249

Page 11 lines 350 352 “Note that the DoF with the highest error were those of the index finger. The lowest error was for flexion/extension of the PIP of the middle finger.” Since only average values of RMSE are considered, these considerations are just qualitative. An ANOVA test should be considered for the evaluation of the highest and the lowest errors

 Page 12. Figure 8. Similar consideration of the previous comment. A statistical analysis of the results of the RMSE should be added for the assessment of the lower error.

RESPONSE: In our opinion, it is no point to perform an ANOVA. The aim of these results is to show the level of magnitude of the errors when estimating angles from other joints. There is no point to show whether the errors of some joints are statistically significant or not from the errors of other joints. The point is: an error of 5deg or 10deg is low enough for some fields (e.g. animation) or activities. In this case, less sensors may be used to record hand kinematics. This is why we only highlighted which joints and/or actions are better or worse predicted.

To improve the readability of the results, a full description of the acronyms should be added in all the capitation of both figures and tables.

RESPONSE: To improve the readability of the results and to not overload captions of figures and tables, the full description of the acronyms has been added in a new section called “Abbreviations”.

Reviewer 3 Report

The paper presents a Synergy-based sensor reduction for recording the whole hand kinematics.

The paper is very well structured and the conclusion are supported by the results.

In my opinion there only Section that have to be improved since the abbreviations are not very well introduced, please clarify all the abbreviation letting the reader better understandthe contents.

After this minor revision in my opinion the paper can be accepted since the methodology is very promising.

Author Response

Reviewer #3 (Remarks to the Authors):

The paper presents a Synergy-based sensor reduction for recording the whole hand kinematics.

 The paper is very well structured and the conclusion are supported by the results.

In my opinion there only Section that have to be improved since the abbreviations are not very well introduced, please clarify all the abbreviation letting the reader better understand the contents.

 After this minor revision in my opinion the paper can be accepted since the methodology is very promising.

 RESPONSE: Many thanks to the reviewer for the suggestion. A full description of the abbreviations and acronyms has been added in a new section called “Abbreviations”.

Round 2

Reviewer 2 Report

the authors have responded to all my requests